# Effectiveness of Virtual Reality on Rehabilitation of Chronic Non-Specific Low Back Pain Patients

**DOI:** 10.3390/healthcare12131312

**Published:** 2024-06-30

**Authors:** Hisham Hussein, Mohamed Atteya, Ehab Kamel

**Affiliations:** 1Department of Physical Therapy, College of Applied Medical Sciences, University of Ha’il, Ha’il 55476, Saudi Arabia; 2Department of Basic Sciences for Physical Therapy, Faculty of Physical Therapy, Cairo University, Giza 12613, Egypt; 3Department of Public Health, College of Public Health and Health Informatics, University of Ha’íl, Ha’íl 55476, Saudi Arabia

**Keywords:** Biodex, musculoskeletal, rehabilitation, spine, video games

## Abstract

Background: Virtual reality (VR) is used extensively for musculoskeletal conditions, but its efficacy in chronic low back pain still needs more investigation. Objective: To discuss the effectiveness of VR on selected outcomes in Chronic Non-Specific Low Back Pain (CNSLBP). Methods: Thirty-five patients with CNSLBP joined this study. Postural correction exercises using the TBed VR gaming system in addition to hamstring stretching were employed, and moist heat on the low back was applied. Pre- and post-intervention values of pain, ROM, function, and balance (overall stability index) were obtained using the numerical rating pain scale (NPRS), Oswestry Disability Index, back range of motion (BROM), and Biodex system. Satisfaction level on a 1–10 scale and the degree of commitment to the exercise sessions were assessed after the intervention. Results: The patients completed the intervention period and outcome measures sessions. Paired t-tests reported statistically significant improvements and high effect size in pain, ROM, function, and balance after the end of the treatment (*p* < 0.001, Cohen’s *d* > 0.69). The level of satisfaction was 9.25 ± 0.766, and the commitment to exercise sessions was high (98.75% attendance rate). Conclusions: Applying postural correction using TBed VR gaming in addition to heat and stretching may improve pain, range of motion, function, and balance in patients with chronic low back pain.

## 1. Introduction

Low back pain (LBP) is defined as discomfort in the lower back and is considered chronic if it has existed for more than three months [1]. Chronic Non-Specific Low Back Pain (CNSLBP) is a type of LBP that lacks clues about underlying pathological conditions. This type of pain can lead to mechanical deformation and can cause dysfunction and disability [2,3].

Treatment of LBP includes many methods from invasive and non-invasive therapies that vary according to various situations. Conservative (non-invasive) treatment may vary between medications, physical therapy, and exercises such as postural correction, strengthening, stretching, and coordination which have been studied extensively and showed a strong level of evidence in reducing pain and enhancing function [4].

On the other hand, physical exercises can be resisted due to a lack of motivation and low adherence [5]. Inactivity and avoidance of exercise are impacted by many factors as physical, environmental, psychological, and sociocultural factors [6,7]. To some extent, how individuals feel during exercise, the happiness they experience, their experience of past activities, and convictions about exercise might be key factors in practice aversion [8,9]. Critically, how individuals feel during exercise predicts their future commitment to activities [9].

To some extent, the enhancement of exercise experiences in people may lead to important effects on their behavior during future exercise practice. Negative beliefs and thoughts experienced by some persons might cause avoidance behaviors due to feelings of pain, leading to inactivity and skipping of therapy sessions. So, these lead to recovery delay and a low level of rehabilitation success [10].

These problems can be faced by using recent gaming and virtual reality (VR) technology, which adds some benefits to traditional rehabilitation activities as it enhances the motivational components and the interactivity of the therapy [11]. The recovery will be conducted in an attractive way where there is active audiovisual feedback for the patient, who is suspended by the environment, leading to increased exercise adherence [12,13].

Gaming is a strong aggravation interruption component on pain as it focuses on the surrounding environment (external stimulus) and not on the movement of the body; this interruption will reduce pain by increasing attention on dividing tasks [13,14]. In addition, gaming is thought of as an efficient tool and is cost-effective when compared to traditional methods [15].

TBed is a four-section treatment couch which is completely sensorized. This couch is provided by a sensor network, which can detect any amount of pressure exerted by the patient’s body and can reflect it in a real-time game on a computer screen attached to the device. Through this screen, the patient can play different games using the contraction of the neck, upper, and lower back muscles which makes it a good choice for video gaming rehabilitation of spinal conditions.

Various studies have discussed the efficacy of gaming and VR in the treatment of LBP [16,17], but still, there is controversy on its short-term effects. In addition, no one has focused on the level of satisfaction and adherence to exercise sessions. Thus, this study aimed to investigate the effectiveness of applying postural correction exercises using VR videogaming on pain, range of motion (ROM), function, and balance in patients with CNLBP. Additionally, this study aimed to assess the level of satisfaction and commitment to exercise sessions when VR was added to the program.

## 2. Materials and Methods

This study was conducted—between March and June 2024—at the Physical Therapy Department at a local university. We recruited patients attending the university clinics and the employees and students of the university. Ethical approval of the study methodology was granted by the university ethical committee board (No: H-2024-370). Additionally, the procedure of this study was explained to the patients, and they gave their signed informed consent before enrolment.

### 2.1. Design

This study employed a pre–post-test design.

### 2.2. Patient Participation

Forty-two subjects with low back dysfunctions were screened; of them, thirty-five patients with CNSLBP were selected to participate in this study. The inclusion criteria were age between 20 and 45 years old, pain in the low back area for more than three months, and pain intensity between 2 and 6 on the numerical rating pain scale (NPRS). The exclusion criteria were severe pain, specific LBP, neurological disorders, psychiatric disorders, spinal surgery, diabetes mellites, and physiotherapy during the last 6 months.

### 2.3. Outcome Measurements

The outcome measures—except commitment and the level of satisfaction—were assessed before the beginning of this study (at baseline) and post-intervention (after four weeks). Commitment and level of satisfaction were assessed once at the end of this study.

#### 2.3.1. Pain Intensity

The NPRS is a scale ranging from 0 to 10, where 0 means no pain and 10 represents the worst pain. The patients were asked to circle/choose the number representing the level of pain they felt. A previous study assessed the validity and reliability of the NPRS and reported high values of 0.941 and 0.95, respectively [18].

#### 2.3.2. Active ROM of the Lumbar Spine

Back range of motion (BROM) is a device designed to assess the ROM of the lumbosacral region. It consists of a set of inclinometers—attached to two plastic frames—arranged in vertical and horizontal fashions to assess the sagittal, frontal, and rotational movement of the lower back. BROM is a valid and reliable device for measuring lumbar ROM [19]. The current study used BROM (Performance Attainment Associates, Roseville, Minnesota) to assess active flexion and extension of the lumbar spine. Measurements were taken according to the guidelines described previously [20].

The AROM was determined by calculating the difference between the angle recorded in the starting position and the angle recorded at the end position of either flexion or extension.

#### 2.3.3. Function

The Oswestry Disability Index (ODI) was used to assess functional disability levels. It consists of 10 sections covering the discomfort of the back and the activities conducted every day. Each item has six questions with a score ranging from 0 to 5, according to the arrangement of the statement, as the first refers to 0 and the second takes 1, etc. Then, the scores are added to calculate the total score which ranges from 0 to 50. This score can be used as raw data, or it can be used to calculate a percentage. In either case, lower values indicate better function and higher values indicate more disability. In the current study, the raw data were used to conduct the analysis, as the raw data were considered more sensitive and thus able to show minor changes compared to ratio values [21]. The validity and reliability of the ODI have been studied and showed a high level in terms of ICC for the reliability range (0.96–0.88) [22].

#### 2.3.4. Balance

The Biodex System (SD 950-340, Biodex Medical Systems, Inc., Shirley, NY, USA) was used to measure the balance as indicated by the overall stability index (OSI). The Biodex device has a multiaxial platform that measures the ability of the individual to be balanced when moving the platform.

The measuring procedures started after entering the patient’s demographic data into the device software; then, the test parameters were selected. In the current study, we used the same parameters conducted in a previous work [23] where the patients were barefoot, standing on both feet; the difficulty level was medium (level 5 with open eyes), each trail was 30 s, and each rest interval was 10 s with one familiarization trial performed before each test. The test procedures require the patient to keep watching the screen, while trying to control the cursor using visio-motor control. Once the test started, five-second delays were counted; then, the platform was released, and the patient tried to maintain the cursor on the center point of the screen.

The OSI is the best indicator of the overall ability of the patient to maintain balance of the platform. It has been designed to calculate the amount of deviation from the baseline position. So, low postural stability is represented as higher OSI scores. The device shows a high level of validity and reliability in measuring balance [24].

#### 2.3.5. Satisfaction Level

This outcome was measured using a 1–10 scale which resembles the NPRS. However, 0 points indicate no satisfaction, while the number 10 indicates maximum satisfaction [25]. The patient was asked to answer the following question: “To what extent can you rate your satisfaction with the introduced physical therapy program?”. The patient then chose any number from 0 to 10 that represented the current level of satisfaction.

#### 2.3.6. Commitment to Exercise Sessions

This outcome was assessed by calculating the percentage (%) of the successful sessions that were attended by the patients, which is subdivided by 420 (the total number of sessions) and then multiplied by 100.

### 2.4. Interventions

The intervention program in this study was designed to consist of twelve sessions, conducted three times per week for four weeks. Each session lasted 40 min on average.

#### 2.4.1. Postural Correction with Lumbar Stabilization

These exercises consisted of pelvic rocking and activation of the upper back and interscapular muscles [26]. A TBed from Techno body©, Dalmine, Italy, was used to assist the patient to perform these exercises through a VR video gaming interface. The patient was asked to contract the targeted muscles and push down on the surface of the TBed. The patient’s pressure activates the sensors which was reflected on the game as triggering of the gun that targets flying fruits. The game consists of three levels: easy, medium, and hard. The patient was asked to play the game with the lower back, where pushing the lower back against the bed (posterior pelvic tilting) triggered the gun, and with the upper back, where pushing the scapula against the bed was the triggering action.

A single familiarization trial followed by three actual trials for the lower back and three other actual trials for the upper back were performed per session. The patient started from the easy level and upgraded to the more difficult levels upon mastering the current level and reaching high scores. The patient started gaming while assuming a supine position. From the 3rd week, a semi-reclined position was used (Figure 1).

#### 2.4.2. Moist Heat

A hot pack of suitable size was used for twenty minutes to apply superficial moist heat. To keep the temperature constant, the hot pack was replaced with a new one after 10 min. The hot pack was wrapped in four layers of towels to keep an appropriate distance between the pack and the skin. The sensation should be no more than moderate warmth. A thermal sensation test using two test tubes was performed before applying heat therapy to avoid any adverse effects [27].

#### 2.4.3. Hamstring and Back Muscle Stretching

Patients received manual passive stretches for the hamstring and lower back muscles. Three repetitions were performed per session, with every repetition sustained for 30 s. The patient assumed a long sitting position with both knees in full extension and the feet together. The patient was then asked to bend forward from the hips to reach the furthest point towards the feet. The therapist applied over pressure by pressing his hands on the patient’s upper back and pushing forward [28].

### 2.5. Statistical Design

Patients’ characteristics were expressed using mean and standard deviation, and homogeneity was assessed by Levene’s test. The mean and standard deviation were calculated for all outcomes. The data were normally distributed as per Shapiro–Wilk test. SPSS (IBM SPSS Statistics, V 27) software was used to conduct paired *t*-tests to determine the within-group differences. The significance level was set at α ≤ 0.05.

## 3. Results

The current study recruited thirty-five patients with CNSLBP. Both sexes participated in this study, with 22 males representing 62.8% and 13 females representing 37.2%. The mean ± SD of the age, weight, height, and BMI of the patients were 43.5 ± 4.76 years, 69.35 ± 7.20 kg, 1.72 ± 0.06 m, and 27.27 w/m^2^, respectively.

The paired t-test statistics reported statistically significant differences in pain intensity, ROM, function, and OSI. This was associated with a high effect size as calculated by Cohen’s *d* formula (Table 1). Additionally, the mean difference reported between the pre- and post-intervention values of pain intensity and ROM were higher than the minimal clinical important difference (MCID) that has been calculated in previous studies.

The average reported level of patients’ satisfaction was 9.25 ± 0.766 where fourteen patients out of thirty-five reported a maximum level of satisfaction of 10/10, seventeen patients reported 9/10, three patients reported 8/10, and a single patient reported 7/10. Regarding the commitment to the treatment sessions, 415 out of 420 sessions were conducted successfully with a commitment percentage of 98.8% of the sessions achieved.

## 4. Discussion

This study was conducted to explore the improvement in selected outcomes in patients with chronic LBP after introducing VR video gaming activity into their physical therapy rehabilitation program. It also investigated the level of commitment and satisfaction after the end of the intervention period. We found significant improvement in all outcomes, namely, pain, ROM, function, and OSI. Additionally, the reported percentages of commitment and satisfaction were high.

To the authors’ knowledge, the use of TBed technology to provide a VR means of helping patients perform exercises for the spine is scarce, so a comparison with the previous literature could be hard. On the other hand, various VR means were utilized by different researchers. For example, Yalfani and colleagues used the HTC Vive virtual reality system—that consisted of headset device and provides several games—with a group of elderly women with chronic LBP and assessed postural sway and physical function. Compared to the control group that received standard exercises, the VR group exhibited a significant effect on postural sway and physical function [29]. In an earlier study, Yalfani’s research group conducted a randomized controlled trial to assess the use of the previously mentioned VR device in the rehabilitation of elderly women with chronic LBP; in this experiment, pain intensity, risk of fall, and quality of life were the outcome measures. Similar to their recent study, they reported significant improvement in all outcome measures [30]. In another study, Matheve and colleagues investigated the distracting effect of VR on the sense and recognition of chronic LBP. They applied a single non-immersive VR session using a wireless motion sensor system (Valedo^®^Pro, Hocoma, Switzerland) to assess pain-related measures during and immediately after the task. They reported that VR effectively induced hypoalgesia during and immediately after the activity and reduced the time spent thinking about the pain [14]. It is worth noting that Matheve’s study used non-immersive method of VR as the current study did. However, the durations of the session and the entire program were shorter than those the current study.

Similarly, Afzal et al. reported significant improvement in pain intensity and functional level in a group of patients after adding VR training to conventional physical therapy while the control group in this study received conventional physical therapy only [31].

As mentioned earlier, TBed technology was not used previously as a source of VR-augmented exercises on patients with chronic LBP. Additionally, the level of satisfaction as reported by the patients and the level of commitment to the rehabilitation program were not assessed in previous studies [32], although both outcomes are closely related to the use of VR. The playful and exciting nature of using VR and gaming could attract patients and improve their attendance at the session, which could also increase their motivation and satisfaction with the introduced service [33] as we reported in the current study.

The improvement reported in the current study can be attributed to several mechanisms; the pain-relieving effect of VR has been reported in previous work [34]. The pain-relieving effect is one of the primary aims of using VR, especially in pediatric burn patients. According to Hoffman and colleagues, analgesic changes in the brain were determined after the application of VR [35]. Additionally, the enjoyment of using VR gaming can help release dopamine, increase feelings of happiness, and mask pain sensations [36]. Pain reduction can indirectly improve patients’ engagement in exercises and improve their participation in the exercise which in turn can improve the outcome. Moreover, the distraction provided by VR gaming might help the patient to overcome the fear avoidance behavior that has been demonstrated by patients with CNLBP.

### 4.1. Implication for Clinical Practice

In light of the current findings, therapists might incorporate VR, especially through the use of TBed, in their rehabilitation of chronic LBP. The reported satisfaction and high commitment to the exercise sessions might be a good advantage and increase the potential success of the treatment. TBed provides gradual difficulty progression in the games; this can be used as a progression of the exercises and allow the patient to exercise different spinal levels (cervical, upper, and lower back) and can also be adjusted so that the exercises can be applied from a supine to long sitting position according to the needs of the patient.

### 4.2. Implication for Future Research

Future researchers need to pay attention to the advantages of the TBed device and examine its various capabilities, especially in the exercise of the spine. Well-designed randomized trials should be conducted to compare its effects with other VR devices. Additionally, the measures of patient satisfaction and commitment to the exercise program should be compared when VR versus regular interventions are conducted.

### 4.3. Limitations

The current study’s limitations can be listed in the following points: Firstly, the lack of a control group may affect the results and limit its generalizability. Secondly, the sample size was not calculated before this study, which makes the sample representativeness questionable. So, the future work should be based on a randomized controlled trial design with an appropriate sample size to obtain more objective findings. Thirdly, subgroups of chronic LBP patients were not considered in this study. Fourthly, the use of heat and stretching in addition to the VR might interfere with the effect that might be gained by the sole use of VR. Fifthly, the current study investigated the short-term effects of using TBed VR gaming. The long-term effects of using VR gaming need to be explored in future research. Finally, the level of satisfaction was assessed using a broad term which might not be closely related to the VR experience; so, we recommend using a more structural questionnaire to assess satisfaction in future work.

## 5. Conclusions

Applying postural correction using TBed VR gaming in addition to heat and stretching may improve pain, range of motion, function, and balance in patients with chronic low back pain.

## Figures and Tables

**Figure 1 healthcare-12-01312-f001:**
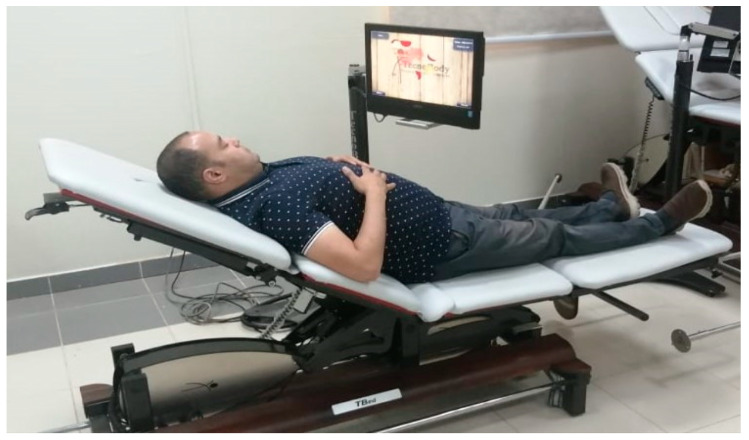
Application of postural correction exercises using TBed.

**Table 1 healthcare-12-01312-t001:** Pre- vs. post-intervention values for pain, ROM, function, and stability index.

Variable	Pre-Intervention	Post-Intervention	MD	95% CI		*t*	*p*	*d*
	M ± SD	M ± SD		Lower	Higher			
Pain intensity by NPRS	4.95 ± 0.88	2.75 ± 1.11	2.20	1.70	2.69	9.31	<0.001	1.05
FROM	41.05 ± 6.62	50.40 ± 8.15	9.35	−12.06	−6.63	−7.21	<0.001	5.79
EROM	10.35 ± 2.36	16.05 ± 1.82	5.70	−6.85	−4.54	−10.30	<0.001	2.47
Function by (ODI)	19.55 ± 5.57	10.85 ± 3.08	8.70	5.95	11.44	6.63	<0.001	5.86
OSI	1.52 ± 0.94	0.80 ± 0.65	0.72	0.40	1.04	4.75	<0.001	0.69

NPRS, numeric pain rating scale; FROM, flexion range of motion; EROM, extension range of motion; ODI, Oswestry Disability Index; OSI, overall stability index; M, mean; SD, standard deviation; MD, mean difference; *t*, *t* value; CI, confidence interval; *p*, significance; *d*, Cohen’s d value.

## Data Availability

The research data will be available from the funding institute and can be provided upon request.

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
