# Peer review of "Effectiveness of Virtual Reality on Rehabilitation of Chronic Non-Specific Low Back Pain Patients"

_healthcare, 2024, doi:10.3390/healthcare12131312_

Round 1

Reviewer 1 Report

Comments and Suggestions for Authors

The manuscript entitled "Efficacy of Virtual Reality on the Rehabilitation of Chronic Non-Specific Low Back Pain Patient: Quasi-Experimental Study" analyses the effects of VR on the Rehabilitation of NSLBP.

Given the significant increase in the use of VR, the application of VR in pain recovery is an essential and timely development.

Please ensure that the manuscript includes suggestions, corrections, or queries to be carried out/answered that could enhance its impact and relevance to the readers.

1.      Typos – The phrases in the abstract end with semicolons, instead they can be completed with full stop punctuation.

2.      TBed can be represented along with its expanded form.

3.      CNSLBP has been abbreviated already in the abstract and does not need to be abbreviated in the introduction section.

4.      Detailing the pain control/modulation mechanism by VR can be included.

5.      The manuscript was submitted to the journal in June 2024, but in the M&M section, it has been specified that the work was conducted between March and July 2024.

6.      The Institutional Ethical Committee number is missing in the manuscript.

7.      Do the exclusion criteria have Diabetes Mellitus patients? DM and LBP have been highly associated.

8.      Treatment done in TBed with VR gaming can be included in the manuscript which might bring more authenticity to the study.

In the result section, scores of 14 and 17 out of 35 samples have been mentioned and the remaining score of 4 samples can be included with its value to support the average of 9.25/10.

Author Response

  1. Typos – The phrases in the abstract end with semicolons, instead they can be completed with full stop punctuation.

Authors’ Reply:

Semicolons used in the abstract were per the journal template, we changed it and used full stop as the reviewer recommend.

  1. TBed can be represented along with its expanded form.

Authors’ Reply:

The term TBed, is the official full name that has been used by the manufacturer in their official website. Please refer to the following link “ https://www.medical-xprt.com/downloads/tecnobody-model-tbed-therapy-and-physiotherapy-cribs-bed-brochure-1081108 ”

  1. CNSLBP has been abbreviated already in the abstract and does not need to be abbreviated in the introduction section.

Authors’ Reply:

If you allow us, the authors believe that the Abstract and the main text are separate entities. So, the abbreviations guideline should be separated in both of them. This can be seen in so many published articles.

These are example from the same journal and latest published articles  

https://doi.org/10.3390/healthcare12131241

https://doi.org/10.3390/healthcare12131240

  1. Detailing the pain control/modulation mechanism by VR can be included.

Authors’ Reply:

Thanks for this comment, we added a complete paragraph describing the possible effects of VR with a focus on the pain-reliving effect. Please refer to lines 255-264

  1. The manuscript was submitted to the journal in June 2024, but in the M&M section, it has been specified that the work was conducted between March and July 2024.

Authors’ Reply:

Thank you for this comment, it should ne June instead of July. We changed it. Please refer to line 78

  1. The Institutional Ethical Committee number is missing in the manuscript.

Authors’ Reply:

Done. The number was added in the main text and before the references. please refer to lines 82 and 306-307

  1. Do the exclusion criteria have Diabetes Mellitus patients? DM and LBP have been highly associated.

Authors’ Reply:

Thanks for this observation. Yes, we excluded any patient with diagnosed DM. so we added it in the exclusion criteria. Please refer to line 92

  1. Treatment done in TBed with VR gaming can be included in the manuscript which might bring more authenticity to the study.

Authors’ Reply:

Done, the detailed procedures and the explanation of the game along with the required tasks from the patients were reported in lines 158-170, and a figure was added.

In the result section, scores of 14 and 17 out of 35 samples have been mentioned and the remaining score of 4 samples can be included with its value to support the average of 9.25/10.

Authors’ Reply:

Required data were added. Please refer to lines 210-213

Reviewer 2 Report

Comments and Suggestions for Authors

The authors have tried to investgate the effectiveness of VR technology in patients with non-spesific back pain, and to present new data regarding satisfaction and adherence.

1. All though the authors state that it is a quasi experimental study, the absence of a control group doesn't apply to it. The study would be more appropriate to be called a pre-post. The absence of a control group also limits generalization of the findings and authors should n't be so certain on them (in the abstracts conclusion they seem certain as they state that " VR gaming 25 can improve pain, range of motion, function, and balance in patients with chronic low back pain".

2.It would add a lot to the manuscript the inclution of pictures of the intervention. 

3. The authors need to present the equipment used for the interface between the patient and the program? Did they use head mounted display or a computer -tv screen? The level of immersion is a significant component that should be described.

4.. the authors used parametric tests, had their data normal distribution? 

5. The investigation of satisfaction seems quite general. The authors didn't use a stractured questionnaire for VR interventions and just asked their patient how they find the treatment in general at the end. A lot of information could be missed. They should better describe what they want to investigate in terms of "satisfaction"

6. Ιn the discussion, line 213, the authors state that the VR was added to the physical rehabilitation program. Heat and stretching could not be a full physical rehabilitation program for this pathology. Please add all components of the rehabilitation program.

7. In the discussion, the authors need to provide further informtion on the vr-games and equipment used in previous RCTs in order to have a clear picture of the difference with this new equipment that they have used.

8.. Clinical implications could npt derive by just a study without control group.

9. Coclusions can't be so certain on the effectiveness. Please rephrase.

10. The population is not hard to find and it would strengthen their work the inclusion of a control group.

Author Response

The authors have tried to investgate the effectiveness of VR technology in patients with non-spesific back pain, and to present new data regarding satisfaction and adherence.

  1. All though the authors state that it is a quasi experimental study, the absence of a control group doesn't apply to it. The study would be more appropriate to be called a pre-post. The absence of a control group also limits generalization of the findings and authors should n't be so certain on them (in the abstracts conclusion they seem certain as they state that " VR gaming 25 can improve pain, range of motion, function, and balance in patients with chronic low back pain".

Authors’ Reply:

Thank you. The design of the study was changes according to the comment.

The level of certainty was reduced in the conclusion of the abstract and in the main text

2.It would add a lot to the manuscript the inclution of pictures of the intervention. 

Authors’ Reply:

Thank you. Picture added. Please refer to line 172

  1. The authors need to present the equipment used for the interface between the patient and the program? Did they use head mounted display or a computer -tv screen? The level of immersion is a significant component that should be described.

Authors’ Reply:

The TBed device provides a disc top screen through which the patient can play. This is described in line 66-69

4.. the authors used parametric tests, had their data normal distribution? 

Authors’ Reply:

Yes, data were normally distributed. Relative data were added in the statistical design section

  1. The investigation of satisfaction seems quite general. The authors didn't use a stractured questionnaire for VR interventions and just asked their patient how they find the treatment in general at the end. A lot of information could be missed. They should better describe what they want to investigate in terms of "satisfaction"

Authors’ Reply:

Authors appreciate this valuable comment

In fact, we agree that we applied general satisfaction measure. But the results might be closely related to the use of VR because the other intervention were kept at minimal.

However, we stated this issue in the limitation section and recommended the use of a structural questionnaire in future research.

  1. Ιn the discussion, line 213, the authors state that the VR was added to the physical rehabilitation program. Heat and stretching could not be a full physical rehabilitation program for this pathology. Please add all components of the rehabilitation program.

Authors’ Reply:

Thank you for this comment,

Authors added this limited number of interventions on purpose because there are no control group we cannot use a full physical therapy intervention in addition to the VR. Instead we preferred to use limited number of interventions that cannot alone provide satisfactory improvement, so the addition of postural correction exercises using VR will be the cornerstone in this program.

  1. In the discussion, the authors need to provide further informtion on the vr-games and equipment used in previous RCTs in order to have a clear picture of the difference with this new equipment that they have used.

Authors’ Reply:

Done, information about the VR devices and types used by other researchers were added. Please refer to lines 226, 227, 231, 236, 237

8.. Clinical implications could not derive by just a study without control group.

Authors’ Reply:

Authors agree. But in light of the current finding we think it’s better to give advice about the use of the device so we did not delete the implication but we rephrase it to decrease the level of certainty and importance.

  1. Coclusions can't be so certain on the effectiveness. Please rephrase.

Authors’ Reply:

Done in the abstract and in the main text

  1. The population is not hard to find and it would strengthen their work the inclusion of a control group.

Authors’ Reply:

That is right, but the aim of this study was predetermined within a funded research project that consists of 2 other experiments so we cannot change the design now. But the authors are now constructing a new larger scale RCT on the same topic.

Reviewer 3 Report

Comments and Suggestions for Authors

This exploratory piece of research uses a good research methodology with promising results concerning the combination of several modalities for intervention in subjects presenting with chronic low back pain (CLBP), namely the use of a new technology (TBed – TecnoBody© ) associated to VR as a way to manage the exercises programme (the postural correction exercises).

There are some pitfalls on the research methodology as mentioned by the authors on point 4.3 (line 260). The limitations should be addressed on futures studies that would benefit as well as of a methodology design compatible to an RCT allowing for the extrapolation of the results. Future work should involve larger sample sizes; control and placebo groups and follow-up to assess the effectiveness of the intervention across time.

Author Response

This exploratory piece of research uses a good research methodology with promising results concerning the combination of several modalities for intervention in subjects presenting with chronic low back pain (CLBP), namely the use of a new technology (TBed – TecnoBody© ) associated to VR as a way to manage the exercises programme (the postural correction exercises).

There are some pitfalls on the research methodology as mentioned by the authors on point 4.3 (line 260). The limitations should be addressed on futures studies that would benefit as well as of a methodology design compatible to an RCT allowing for the extrapolation of the results. Future work should involve larger sample sizes; control and placebo groups and follow-up to assess the effectiveness of the intervention across time.

Authors’ Reply:

Thank you

The issues mentioned about the recommendation for future research along with the limitations were addressed in the limitation section.

Reviewer 4 Report

Comments and Suggestions for Authors

Dear authors

Thank you for the manuscript. The work addresses a topic that is not new and that has been extensively studied, however, what seems to me to be intended to be new is the use of virtual reality, which in my opinion is meritorious. Generally speaking, it is well structured, the introduction is robust and methodologically objective and understandable.

My main concerns relate to the choices made in carrying out the work, which I will now explain:

If the objective is to verify the effectiveness of VR in the defined outcomes in chronic low back pain, why opt for a combined intervention (VR+exercises+humid heat+stretching), since in this way and given the design of the study, it will not be possible to attribute the result to virtual reality, as the authors conclude.

In addition, as mentioned in the introduction, for the indicators used, the interventions associated with VR in this study have already proven to be effective.

In this sense, I believe that these concerns of mine should be added to the limitations of the study, and the conclusion should be reformulated to be more precise.

I also believe that the term efficacy should be replaced by effectiveness, since efficacy studies refer to RCTs.

Finally, it is not clear whether the study was approved by the ethics committee. To prove this, the identification referred to in the materials and methods, in line 81, and in line 280, should appear.

In line 136, replace “five five-second…” with “five-seconds”

Author Response

Dear authors

Thank you for the manuscript. The work addresses a topic that is not new and that has been extensively studied, however, what seems to me to be intended to be new is the use of virtual reality, which in my opinion is meritorious. Generally speaking, it is well structured, the introduction is robust and methodologically objective and understandable.

My main concerns relate to the choices made in carrying out the work, which I will now explain:

If the objective is to verify the effectiveness of VR in the defined outcomes in chronic low back pain, why opt for a combined intervention (VR+exercises+humid heat+stretching), since in this way and given the design of the study, it will not be possible to attribute the result to virtual reality, as the authors conclude.

Authors’ Reply:

Thank you for this important comment,

We thought that using heating and stretch which are not a complete session fot these cases may be the appropriate choice. We could not use the TBed for the application of the entire intervention and also, we cannot use postural correction exercise alone.

We addressed this issue in the limitations and highlighted the need for future RCTs with control group to reach more objective results

In addition, as mentioned in the introduction, for the indicators used, the interventions associated with VR in this study have already proven to be effective.

In this sense, I believe that these concerns of mine should be added to the limitations of the study, and the conclusion should be reformulated to be more precise.

Authors’ Reply:

This issue was added to the limitations

The conclusion was paraphrased

I also believe that the term efficacy should be replaced by effectiveness, since efficacy studies refer to RCTs.

Authors’ Reply:

Done, the term efficacy was replaced with effectiveness

Finally, it is not clear whether the study was approved by the ethics committee. To prove this, the identification referred to in the materials and methods, in line 81, and in line 280, should appear.

Authors’ Reply:

Done, the ethical approval number was added in the two places

In line 136, replace “five five-second…” with “five-seconds”

Authors’ Reply:

Done, please refer to line 137

Round 2

Reviewer 2 Report

Comments and Suggestions for Authors

The authors answered to all issues raised and made appropriate corrections in the manuscript. Congratulations.

Reviewer 4 Report

Comments and Suggestions for Authors

Dear Authors

I would like to thank the authors for their efforts in improving the manuscript; the result seems to address my concerns. Congratulations